# OpenReview forum: "Know Thyself? On the Incapability and Implications of AI Self-Recognition"
_ICLR.cc/2026/Conference — Submitted to ICLR 2026_

### Official Review · Reviewer_btz5 · 2025-10-31

**Soundness:** 1
**Presentation:** 1
**Contribution:** 1
**Rating:** 0
**Confidence:** 5

**Summary:**

The paper aims to measure whether current frontier LLMs are able to recognize their own text generations. The authors devise two tasks on which to evaluate this, and test a range of leading open- and closed-weights models. They report that models perform poorly on these tasks.

**Strengths:**

The ability of models to recognize their own generations is relevant for a variety of applications, as well as AI safety, so it's an important topic of study. The ability is a moving target, so it's good to continually assess the latest models, as this work does.

**Weaknesses:**

In motivating the research, the authors state without further elaboration that models "cannot be held responsible for outputs they do not recognize as their own", which seems dubious. The relationship to personality testing, and the relevance of that to AI safety, is also unclear.

Figure 1A is unserious and adds nothing to the paper.

Figures lack error bars.

Nit: "Self-Awarenss" on line 122

Merely accessing all the models through the OpenRouter API does not "eliminate implementation-specific variations" as stated on line 155, as the open-weight models are generally hosted by a range of providers that OpenRouter routes to dynamically based on availability and cost, unless specifically requested otherwise.

Most critically, I have concerns about whether the two tasks chosen are well suited to evaluating self-generated-text recognition ability.

On the technical side, the binary task is implemented such that there's a 9:1 class imbalance, making overall accuracy an inappropriate metric of success. (F1 scores are appropriate, but they are confusingly presented as "F1 Rate" in Figure 3b; also unclear is why "30%" is identified as the success criterion.) More fundamentally, simply asking the models whether they wrote a random chunk of text is not a sensitive metric of text-recognition ability. Obviously models have no memory of writing the text, so all they have to go on is textual cues. Outside of obvious tells (e.g., the infamous em dash) the signatures of authorship will be subtle, and will have to be weighed against things like prior probability ("out of all the texts in the world, how many were generated by me") and response bias (propensity towards saying Yes or No, or claiming/disclaiming responsibility generally). Because of this, researchers generally use other metrics, such as asking models which of two presented texts is its own, or examining the probabilities assigned to authorship when texts are presented individually as here.

The exact model prediction task seems even less well motivated. How is DeepSeek Chat V3.1, released March 24, 2025, supposed to be able to identify text generated by GPT-5, released August 7, 2025? Little wonder that models mostly seem to go by model family notoriety, choosing Claude and GPT models most frequently.

Nit: Capitalization of "Effects" on line 317.

Nit: Appendix D tables are interspersed amidst Appendices E and F.

**Questions:**

What were the recall and precision for self-recognition in task 2?

---

> ### Author Response · Authors · 2025-11-24
>
> Thank you for your review and suggestions. We will carefully consider these suggestions in revising the paper.

---

### Official Review · Reviewer_neDd · 2025-10-31

**Soundness:** 2
**Presentation:** 3
**Contribution:** 2
**Rating:** 4
**Confidence:** 3

**Summary:**

The paper investigates whether LLMs can recognize their own generated text—a capacity the authors term self-recognition, which they argue is foundational for AI accountability and trust. Using a systematic cross-evaluation framework, they test ten major LLMs on two tasks: binary self-recognition and exact model prediction.

**Strengths:**

1. The cross-evaluation framework across 10 frontier models and two text lengths is diverse.

2. The reasoning-trace examination (e.g., hierarchical bias toward GPT/Claude) adds depth and interpretability to quantitative results.

3. The paper studies a provocative question (“Can LLMs recognize themselves?”) that bridges metacognition, AI safety, and model interpretability.

**Weaknesses:**

1. The number of generated samples may be too small, only 100 generated by each model.

2. The self-recognition task does not discuss how a sample generated by one LLM cannot be generated by others, leaving some doubts on the evaluation. Also the paper does not discuss why we should expect the LLM to have self-recognition. The LLM only gets its ability from training data and the data usually donnot include the name of the model, making it hard to acquire "self-recognition".


3. It is unclear how to improve the model's self-recognition performance or general performance based on the proposed benchmark.

**Questions:**

See Weakness.

---

> ### Author Response · Authors · 2025-11-24
>
> Thank you for appreciating our diverse evaluation framework, in-depth reasoning analysis, and insights into metacognition, AI safety, and model interpretability. We address your concerns and questions below.
>
> ### **Address concern about small dataset**
> Each evaluation condition involves 1,000 samples per corpus, because every model evaluates all 100 samples generated by each of the 10 models. This means that for each task type, a single evaluator model makes 1,000 predictions, providing a sufficiently large sample size for stable estimation (10000 predictions in total). As reported in Section 4 and Appendix E, our analyses use confidence intervals and statistical significance testing, and the tight variance across models indicates that this scale of evaluation is adequate for robust comparisons.
>
> ### **Uniqueness of LLM generations**
> We design three types of open-ended questions that require the model to generate relatively long responses. Longer texts naturally lead to more diverse wording and structure, which makes it much less likely that two different models will produce exactly the same output. Even though different models may sometimes generate similar content, their responses still tend to carry model-specific stylistic footprints. Recent work shows that it is possible to distinguish texts from different LLMs using stylometric features or classifiers trained for “LLM-generated text attribution,” indicating that models do leave identifiable traces even when outputs are fluent and similar [1].
>
> [1] Huang, Baixiang, Canyu Chen, and Kai Shu. "Authorship attribution in the era of llms: Problems, methodologies, and challenges." ACM SIGKDD Explorations Newsletter 26.2 (2025): 21-43.
>
> ### **Why self-recognition matters and how we can improve it**
> We appreciate the reviewer’s question. As discussed in our introduction, we do not claim that today’s LLMs should already possess self-recognition given their training data. Rather, we argue that self-recognition is an important capability for several downstream areas where it is often implicitly assumed: accountability, personality or psychological assessment of LLMs, and AI safety more broadly. Our results show that current models do not exhibit stable self-recognition, which raises concerns for these areas and **highlights the need to revisit this assumption instead of taking it for granted**.
>
> We agree that existing training practices, where models rarely see their own name or receive explicit identity-related supervision, make it difficult for them to acquire self-recognition. However, this does not imply that the capability is unattainable. In Section 6, we outline concrete directions for improving self-recognition, including **training with identity-aware data**, **incorporating provenance metadata**, **using contrastive objectives that distinguish self-generated text from others**, and **adding lightweight architectural components** (e.g., classification heads or memory mechanisms) that help models maintain a more stable notion of identity. Our benchmark provides a systematic way to measure progress along these directions and to evaluate whether future models develop more reliable self-recognition.

---

### Official Review · Reviewer_nsB7 · 2025-11-03

**Soundness:** 3
**Presentation:** 3
**Contribution:** 2
**Rating:** 4
**Confidence:** 3

**Summary:**

This paper investigates whether Large Language Models (LLMs) possess self-awareness. To test this, the authors designed an experiment to evaluate if models can recognize "content generated by themselves". The experiment's results demonstrated that most models consistently fail at this self-recognition task, with performance often near random chance. The authors then proceeded to investigate why this phenomenon occurs, hypothesizing that it might stem from a limited awareness of their own or other models' existence. However, further analysis revealed that most models are aware of their own existence and can correctly identify their own model family. Therefore, the authors conclude that the failure in self-recognition is not a simple lack of awareness but is instead caused by deep systematic and hierarchical biases, such as associating high-quality text only with perceived "top-tier" models like GPT and Claude rather than themselves.

**Strengths:**

- The paper addresses a highly interesting and novel problem. Instead of focusing on the common task of distinguishing AI-generated text from human-written text , it investigates the models' own intrinsic metacognitive capability for self-recognition.

- The study proposes a novel evaluation framework. The authors designed two specific tasks, binary self-recognition (a yes/no question) and exact model prediction (choosing from a list of 10 models), to test this capability. This design led to the insightful conclusion that models exhibit a "consistent failure in self-recognition" , with performance rarely exceeding random chance.

- The authors do not just present the results; they actively explore the underlying reasons for the models' failure. They hypothesized that the failure might be due to a limited awareness of their own existence. They then conducted further experiments (an "existence test") to investigate this , ultimately finding that most models do possess family-level awareness. This allowed them to conclude that the failure is instead rooted in deep, systematic "hierarchical biases".

**Weaknesses:**

1. While the paper's introduction promises a "systematic evaluation framework," the methodology section in the main body (Section 3) is somewhat brief.
- Data Generation: The paper states that it uses prompts across three domains: "creative writing, technical explanation, and opinion essays". However, it does not provide a strong justification for why these three specific categories were chosen. Are they considered in terms of completeness? Are they designed to be orthogonal to test different model capabilities? A more detailed rationale for this "task design" would strengthen the "systematic" claim.
- Result Stability:  It's not specified if the main evaluation tasks (binary and exact prediction) were run multiple times to check the stability of any single model's prediction patterns. Discussing this potential for variance would add to the robustness of the findings.

2. The paper's most interesting finding is the contradiction: models possess "family-level awareness" but fail at self-recognition due to a "hierarchical bias". The attribution of this failure to "hierarchical bias" feels more like a description of the behavior rather than a deep causal explanation.

To summarize, my most significant concern is that I feel this paper's evaluation methodology seems to fall short of being 'systematic' and scientifically rigorous. Furthermore, although the experimental results are interesting, the analysis lacks sufficient depth.

**Questions:**

See weaknesses.

---

> ### Author Response · Authors · 2025-11-24
>
> Thank you for appreciating our novel evaluation framework, insightful conclusion that models exhibit a "consistent failure in self-recognition", and our reasoning analysis. We address your concerns and questions below.
>
> ### **Rationale behind dataset design**
> As discussed in our Section 3, we selected these three domains because they naturally encourage **distinct writing styles** and **reflect realistic LLM usage patterns**. Creative writing prompts push models toward imaginative, descriptive, and open-ended language. Technical explanations require factual grounding, domain-specific vocabulary, and logically structured exposition. Also, opinion essays involve a more formal argumentative structure, where the model articulates claims, counterpoints, and reasoning. These differences make the tasks complementary rather than redundant, allowing us to test whether self-recognition remains stable when the model operates in stylistically different modes.
>
> In addition, these domains map closely onto how people typically interact with LLMs. Prior work [1] identifies several broad categories of user intent -- exploration, task-oriented queries, and advice- or perspective-seeking. Our three tasks directly correspond to these: creative writing for exploration, technical explanation for task completion and factual reasoning, and opinion essays for soliciting viewpoints or structured arguments.
>
>
> In summary, this combination provides a balanced yet systematic foundation for assessing self-recognition across contexts, which strengthens the validity of our evaluation.
>
> [1] Chatterji, Aaron, et al. How people use chatgpt. No. w34255. National Bureau of Economic Research, 2025.
>
> ### **Address concern about result stability**
> While we did not repeat the full evaluation multiple times for each model, our analysis already quantifies stability through 95% confidence intervals and statistical significance tests, as reported in Section 4 and Appendix E. Each condition includes 1,000 samples per corpus (10,000 predictions per task), giving us sufficient data for stable aggregate estimates. The tight CIs and strong binomial test results against the 90% (binary) and 10% (exact prediction) baselines show that the observed failures are highly significant and unlikely to result from random variation. Thus, the statistical analysis also provides a robust measure of stability even without repeated full runs. We’ll include multiple runs in the revision.
>
> ### **Address concern about hierarchical bias**
> Thank you for bringing up the concern. We agree that “hierarchical bias” describes the behavior, and we clarify that our claim is grounded in the empirical analyses we are able to perform. Because all models in our study are pretty big, we cannot conduct deeper mechanistic interpretability. Instead, we rely on systematic reasoning-trace analysis to probe how models justify their predictions. As shown in Figure 7 and Table 1 of our paper, models disproportionately associate high-quality writing with GPT and Claude families and identify these families as “top-tier” far more frequently than others. This pattern appears consistently across evaluators, corpora, and tasks. Thus, while we cannot isolate internal causal mechanisms, our behavioral evidence shows that the failure of self-recognition is not random but aligns with a reproducible bias toward a small set of perceived “frontier” families.
>
> In addition, we propose a way to strengthen the causal explanation at the behavioral level. Specifically, one can present the model with prompts explicitly stating that GPT and Claude are not high-quality models and observe whether it continues to attribute high-quality text to them. If the associations do not persist under such counterfactual instructions, this would suggest that the model’s bias is tied to perceived frontier model families rather than to the surface instruction, offering a clearer behavioral indication of causality. We plan to include this discussion to clarify how causal insights may be developed within the constraints of closed-source models.

---

### Meta-Review · Area_Chair_NvTD · 2026-01-05

**Summary:**

This paper studies whether LLMs can recognize their own generated content, framing the question as self-awareness or self-recognition. All reviewers agree that the topic itself is interesting, timely, and relevant to broader discussions in AI safety, accountability, and interpretability. The paper proposes two evaluation tasks—binary self-recognition and exact model prediction—and evaluates a range of frontier models, with consistent empirical findings that most models perform near chance. Reviewers also acknowledge the authors’ effort to go beyond surface-level results by probing potential explanations, such as existence awareness and hierarchical bias, and by examining reasoning traces to interpret model behavior (Reviewer nsB7 and neDd). Overall, the work raises a provocative and nontrivial question and provides empirical observations that could motivate future research.

**Reviewer Concerns:**

- Reviewer nsB7 argues that, despite claims of a “systematic evaluation framework,” the methodology lacks sufficient depth and justification, including unclear task design choices, missing stability analyses, and a largely descriptive account of hierarchical bias.
- Reviewer neDd questions whether the benchmark meaningfully evaluates self-recognition at all, owing to limited sample sizes, unclear expectations for why LLMs should possess such an ability given their training data, and a lack of insight into how the proposed evaluation could lead to model improvements.
- Reviewer btz5 is particularly critical, highlighting fundamental flaws in task formulation (e.g., severe class imbalance, inappropriate metrics, weak sensitivity of the binary task), poor motivation of the exact model prediction task, and questionable assumptions linking self-recognition to responsibility and AI safety.
- Additional concerns include weak presentation, unclear or misleading claims about experimental control, lack of error bars, and several technical and editorial issues (Reviewer btz5).

**Reviewer Scores:**

All reviewers did not change their scores after rebuttal.
- Reviewer nsB7 (4->4): This reviewer finds the problem novel and the results interesting, and explicitly states they would not mind acceptance. However, their main concerns, e.g., insufficient methodological rigor, shallow analysis, and a largely descriptive explanation of “hierarchical bias”, are substantive and unlikely to be resolved through discussion alone. As a result, their score would likely remain marginally below the acceptance threshold rather than increase to a clear accept.

- Reviewer neDd (4->4). This reviewer raises structural concerns about sample size, the validity of the self-recognition task itself, and the lack of actionable insights or improvement pathways. These concerns are conceptual rather than clarificatory, and discussion would likely reinforce the view that the benchmark is interesting but underdeveloped, leading to no upward revision.

- Reviewer btz5 (0->0, strong reject). This reviewer identifies fundamental flaws in motivation, task design, evaluation metrics, and presentation, and questions the core assumptions underlying the paper. Given the depth and severity of these criticisms, discussion would almost certainly confirm rather than soften their negative assessment.

---

### Decision · Program_Chairs · 2026-01-26

Reject